# Ligand-Induced Activation of GPR110 (ADGRF1) to Improve Visual Function Impaired by Optic Nerve Injury

**DOI:** 10.3390/ijms24065340

**Published:** 2023-03-10

**Authors:** Heung-Sun Kwon, Karl Kevala, Haohua Qian, Mones Abu-Asab, Samarjit Patnaik, Juan Marugan, Hee-Yong Kim

**Affiliations:** 1Laboratory of Molecular Signaling, NIAAA, National Institutes of Health, 5625 Fishers Lane Room 3S-02, Rockville, MD 20892-9410, USA; 2Visual Function Core, NEI, National Institutes of Health, Bethesda, MD 20892-0616, USA; 3Electron Microscopy Laboratory, Biological Imaging Core, NEI, National Institutes of Health, Bethesda, MD 20850-2510, USA; 4Division of Pre-Clinical Innovation, NCATS, National Institutes of Health, Rockville, MD 20817, USA

**Keywords:** synaptamide, A8, axonal degeneration, retinal ganglion cells, neuronal survival, optic nerve crush, visual evoked potential

## Abstract

It is extremely difficult to achieve functional recovery after axonal injury in the adult central nervous system. The activation of G-protein coupled receptor 110 (GPR110, ADGRF1) has been shown to stimulate neurite extension in developing neurons and after axonal injury in adult mice. Here, we demonstrate that GPR110 activation partially restores visual function impaired by optic nerve injury in adult mice. Intravitreal injection of GPR110 ligands, synaptamide and its stable analogue dimethylsynaptamide (A8) after optic nerve crush significantly reduced axonal degeneration and improved axonal integrity and visual function in wild-type but not *gpr110* knockout mice. The retina obtained from the injured mice treated with GPR110 ligands also showed a significant reduction in the crush-induced loss of retinal ganglion cells. Our data suggest that targeting GPR110 may be a viable strategy for functional recovery after optic nerve injury.

## 1. Introduction

CNS injuries in adulthood are difficult to repair [1]. In particular, the inherently low axon growth capacity of mature neurons [2] along with neuronal cell death caused by injury [3,4] hinder the development of recovery strategies for functional restoration. Although limited, the regeneration and repair of the optic nerve system have been demonstrated by manipulating genes or signaling pathways to stimulate the intrinsic developmental program for axon growth or to promote neuronal survival [5,6,7,8]. In addition, the induction of cyclic adenosine monophosphate (cAMP) signaling has been proposed to be a promising strategy to stimulate axon growth [9,10,11]. We have demonstrated that *N*-docosahexaenoylethanolamine (synaptamide), an endogenous metabolite of docosahexaenoic acid (DHA, 22:6n-3), potently promotes neurite outgrowth and synaptogenesis in developing neurons by binding to the G-protein coupled receptor 110 (GPR110, ADGRF1) and increasing cAMP [12,13]. Recently, we have demonstrated that this developmental mechanism for neurite outgrowth is applicable to a repair strategy for axonal injury in the adult stage, as GPR110 is upregulated after injury [14,15]. For example, activating GPR110 using its ligands stimulated axonal extension following optic nerve injury [14] and improved the optic nerve axonal pathology and visual dysfunction caused by a traumatic brain injury (TBI) in adult mice [15]. In this study, we investigated whether ligand-induced GPR110 activation can improve visual function and axonal integrity impaired by crush-induced optic nerve injuries in adult mice. 

## 2. Results

### 2.1. Synaptamide Stimulates Axon Regeneration after ONC 

We have previously demonstrated that GPR110 ligands, synaptamide and A8 stimulate axon extension in injured optic nerves [14]. At 3 weeks after ONC, synaptamide treatment dose-dependently increased GAP43 staining, indicating that axon regeneration was stimulated (Figure 1A,B). CTB labeling was detected together with GAP43-positive staining in injured axons when treated with synaptamide, indicating that regenerating axons at least in part contributed to synaptamide-induced axon extension after the injury. The mass spectrometric analysis of synaptamide in the eye indicated that the synaptamide level decreased by 20% from the initially injected amount at 1 h after the injection, and further decreased by nearly 80% at 24 h (Figure 1C). Nevertheless, a single intravitreal injection of synaptamide at 2.5 mg/kg appeared to be effective to trigger axonal regeneration and extension during the subsequent 3 weeks of recovery in the injured animals (Figure 1A, bottom). A more stable and effective ligand, A8 [14,15], also showed a time-dependent decrease, but to a lesser extent. 

### 2.2. GPR110 Activation by Synaptamide or A8 Treatment Leads to Partial Restoration of Visual Activity Impaired by ONC

The functional outcome of GPR110 activation was determined by evaluating the effect of synaptamide and A8 treatment on visually evoked potentials (VEPs) [16]. At 12 weeks after ONC, a drastic reduction in VEP amplitude was observed in the animals treated with the vehicle, indicating that ONC caused vision impairment (Figure 2). Remarkably, a single intravitreal injection of synaptamide (2.5 mg/kg) or A8 (0.03 mg/kg) following injury improved their visual function, showing partial restoration of the VEP amplitude with a normalized VEP wave (Figure 2A). In *gpr110 KO* mice, however, synaptamide or A8 injection did not reverse the reduction in VEP amplitude caused by ONC (Figure 2B), indicating that the observed in vivo effect of synaptamide and A8 on the functional outcome was mediated by GPR110. The improvement of VEP amplitude was not shown when the injured WT mice were treated with OEA (2.5 mg/kg), which was used as a bio-inactive lipid control for synaptamide. The electroretinogram (ERG) of WT and *gpr110 KO* mice indicated no significant effect of either injury or treatment (Figure 2C). 

### 2.3. ONC-Induced Loss of RGC Axons at the Brain Target was Partly Prevented by the Treatment with GPR110 Ligands

The observed partial recovery of VEP function by the treatment with GPR110 ligands suggests that at least some RGC axons successfully achieved or maintained target innervation in the brain. The axon innervation to the lateral geniculate nucleus (LGN), the brain target of RGC axons, was visualized by anterograde CTB labeling. At 12 weeks after injury, the axon labeling at LGN was clearly missing in animals injured by ONC (Figure 3), indicating that ONC caused the degeneration of distal axons. The axon labeling at the target partially reappeared after the treatment with synaptamide or A8 in WT, but this effect was not observed in *gpr110 KO*, which is consistent with the GPR110-dependent partial restoration of visual function observed in Figure 2.

### 2.4. Optic Nerve Myelination Status Degraded after ONC Was Improved by Synaptamide or A8 Treatment 

To determine the integrity of the optic nerve in relation to the functional outcome, the myelination status was evaluated for the optic nerve tissues collected from the animals in which VEP and ERG were measured at the end of the 12-week recovery period (Figure 4). ONC appeared to cause myelin degradation, as the myelin basic protein (MBP) immunostaining in the cross-section of the optic nerve fibers was markedly decreased after ONC (Figure 4A,B). The level of oligodendrocyte marker proteins, MBP, CNPase and O2 in the optic nerve tissues was also significantly decreased after ONC (Figure 4C,D), indicating that ONC caused the degradation of oligodendrocytes that are responsible for the myelination of neuronal axons. The treatment with synaptamide or A8 partly, but significantly, prevented the loss of these proteins at 12 weeks post-injury, indicating that these GPR110 ligands improved the myelination status. 

### 2.5. GPR110 Ligands Improved Structural Integrity of Axons Deteriorated by ONC 

Because myelin degradation has been shown to accompany cytoskeletal disintegration of distal axons after injury [17], we also examined the ultrastructure of the optic nerves by electron microscopy (Figure 5). The electron micrographs of the injured optic nerves showed a severely disorganized structure with a drastically reduced axon number and apparent penetration of astrocytic processes. Synaptamide or A8 treatment significantly increased the total axon numbers in WT, but not in *gpr110 KO* animals (Figure 5A,B). The thickness of the myelin sheath was also significantly decreased after ONC, as indicated by the increased *g*-ratio [18]. However, this increase in the *g*-ratio was reversed by synaptamide or A8 treatment (Figure 5C–E). These data indicated that the structural integrity of axons deteriorated by injury can be reversed by ligand-induced GPR110 activation.

### 2.6. RGC Loss after ONC Was Alleviated by Synaptamide or A8 Treatment

We also examined the surviving RGCs in the retina collected from the animals after VEP and ERG were measured at the end of the 12-week recovery period (Figure 6). The β-III tubulin-positive RGC neurons and neurites significantly decreased after ONC (Figure 6A,B), as has been reported earlier [19,20]. The ONC-induced loss of RGCs and neurites was partially rescued by the intravitreal injection of synaptamide or A8.

## 3. Discussion

In this study, we demonstrated the GPR110-dependent restoration of the visual function impaired after optic nerve injury in adult mice. The treatment with GPR110 ligands after optic nerve crush injury at least partially prevented axonal degeneration and RGC loss, improving visual function.

GPR110, an adhesion GPCR that has recently been deorphanized as the target receptor of synaptamide [12], mediates neurogenic, neuritogenic and synaptogenic activity in developing neurons through the activation of cAMP/PKA signaling [21]. We have recently demonstrated that GPR110 expression in RGCs is rapidly induced after optic nerve injury, and that activating GPR110-mediated cAMP/PKA signaling, a developmental mechanism of stimulated axon growth, enables the extension of injured axons in adulthood [14]. Similarly, it has been previously shown that cAMP/PKA signaling can regulate neuronal regenerative capacity [22], and by priming with neurotrophins, the axon growth inhibitory signals derived from CNS glia around the injury site can be overcome in a cAMP- and PKA-dependent manner [23]. Axon regeneration observed after synaptamide treatment (Figure 1) is consistent with the reported cAMP-dependent regenerative capacity, since synaptamide is an endogenous ligand to GPR110 that activates cAMP/PKA signaling. Considering synaptamide’s potent neuritogenic activity with IC50 at a low nanomolar range [12], a local concentration of synaptamide from a single injection of synaptamide at 2.5 mg/kg may be sufficient to trigger axon repair in the early stage of injury and stimulate axonal regeneration during the subsequent recovery period. The drastic reduction in VEP observed after ONC (Figure 2) was accompanied by the nearly absent optic nerve innervation at LGN, the target area for RGC axons in the brain (Figure 3), together with the significant degeneration of optic nerve axons (Figure 4 and Figure 5). When injured animals were treated with GPR110 ligands, synaptamide or A8, the structural integrity of axons post injury was improved (Figure 4 and Figure 5), and optic nerve axons were detected in LGN (Figure 3) in a GPR110-dependnet manner. The axonal degeneration alleviated through the activation of cAMP/PKA signaling by specifically targeting GPR110 may have facilitated the partial restoration of visual function. Given that GPR110 plays an important role in developmental neurite outgrowth, the lack of GPR110 may have affected RGC axonal outgrowth during development. However, the total RGC axon number per field (Figure 6A,B), as well as the VEP amplitude (Figure 2), appeared to be similar between WT and KO sham animals, suggesting that unlike the injury situation, other mechanisms for axonal outgrowth can compensate in the long run for the developmental deficit caused by the lack of GPR110 activation.

Axonal injury caused by ONC is known to be mild. It has been reported that RGC survival of greater than 40% can be achieved depending on the severity of the crush [24]. It has also been reported that the loss of RGC soma is not required for axon degeneration. We observed more than 10% of the total RGCs were still spared at 12 weeks after injury (Figure 6). It is noteworthy that GPR110 activation by its ligands improved the survival of RGC neurons after the optic nerve injury (Figure 6). In addition to stimulating axon growth after injury, it is possible that GPR110 activation that elevates the cAMP level may have a specific role in survival/death signaling to help preserve injured neurons, especially when target-derived trophic support is absent after ONC. Similarly, it has been reported that cAMP analogues can promote neuronal survival and neurite outgrowth independently of nerve growth factors [25]. The timely repair of injured axons by the activation of GPR110/cAMP signaling may have prevented the death of neurons traumatized by ONC, preserving their capacity to sprout neurites to form new synapses for signal transmission. In such a case, the rapid induction of GPR110 in RGCs after injury that has been previously observed [14] may be an intrinsic mechanism for preserving RGCs and stimulating axon growth through activating GPR110/cAMP signaling. Further studies will be required to fully understand the mechanisms underlying GPR110-derived trophic signals for RCG survival after injury.

The effect of synaptamide and its stable analogue A8 was observed with a single treatment after ONC, suggesting that GPR110 activation in an early stage of injury can mitigate RGC death (Figure 6) and axon degradation (Figure 4 and Figure 5), and thus help preserve visual function (Figure 2). The partial restoration of VEP function observed in the present study (Figure 2) is a rare demonstration of successful functional recovery after CNS injury. After spinal cord injury, improvement of motor function has been demonstrated with pharmacologic intervention to decrease scarring using chondritinase ABC [26] or the microtubule-stabilizing drug epothilone B [27]. Although clinically inapplicable, combination treatment with the inflammation-inducer zymosan and a cAMP analog, along with *pten* deletion to reactivate the growth potential, was also shown to trigger partial restoration of functional responses after optic nerve injury [20]. To the best of our knowledge, the pharmacologic activation of GPR110 by its ligands represents the first demonstration of functional improvement with translational potential following optic nerve injury. It is also noteworthy that the activation of GPR110/cAMP signaling has been shown to suppress neuroinflammation caused by traumatic brain injury [15] or endotoxin administration [28], indicating the possibility that these GPR110 ligands may be similarly effective in ameliorating inflammation-associated neuropathological conditions such as ischemia and Alzheimer’s disease. Further investigation is warranted to find an optimal and practical therapeutic time window for a treatment strategy based on ligand-induced GPR110 activation. 

In summary, our data indicate that GPR110 activation by synaptamide or its stable analogue A8 promotes RGC survival after optic nerve injury, preserves axonal integrity and enables at least partial recovery of visual function. Our findings demonstrate a novel restoration strategy for injured axons by activating GPR110 using its ligands synaptamide or A8. We propose that synaptamide and its stable analogue may have therapeutic potential for patients who have suffered an optic nerve injury, offering a new translational possibility for CNS injuries. 

## 4. Materials and Methods

### 4.1. Animals

Timed pregnant female C57BL/6 mice were obtained from the NIH-NCI animal production program or Charles River Laboratories (Portage, MI, USA), and GPR110 (adhesion G protein-coupled receptor F1: Adgrf1) heterozygous mice with a C57BL/6 background were generated by the Knockout Mouse Project (KOMP) Repository. The animals were housed in the SPF facility and *gpr110* KO mice and matching WT were generated by heterozygote mating in the NIAAA animal facility. All experiments were carried out in accordance with the guiding principles for the care and use of animals approved by the National Institute on Alcohol Abuse and Alcoholism (LMS-HK13).

### 4.2. Optic Nerve Crush (ONC)

Mice at 2 months of age were anesthetized by an injection of ketamine (100 mg/kg, i.p.) and xylazine (10 mg/kg, i.p.). Under a binocular operating scope, a small incision was made with spring scissors (cat. #RS-5619; Roboz, Gaithersburg, MD, USA) in the conjunctiva beginning inferior to the globe and around the eye temporally. Caution was taken, as making this cut too deep can result in cutting into the underlying musculature (inferior oblique, inferior rectus muscles or the lateral rectus) or the supplying vasculature. With micro-forceps (Dumont #5/45 forceps, cat. #RS-5005; Roboz), the edge of the conjunctiva next to the globe was grasped and retracted, rotating the globe nasally, which exposed the posterior aspect of the globe, allowing visualization of the optic nerve. The exposed optic nerve was grasped approximately 1–3 mm from the globe with Dumont #N7 cross-action forceps (cat. #RS-5027; Roboz) for 5 s to apply pressure on the nerve by the self-clamping action. The Dumont cross-action forceps were chosen because their spring action applied a constant and consistent force to the optic nerve. During the 5 s clamping, we were able to observe mydriasis. After 5 s, the pressure on the optic nerve was released and the forceps removed, allowing the eye to rotate back into place. The ONC operation was performed on one eye for each mouse. 

### 4.3. Intravitreal Administration of Synaptamide or A8

Synaptamide (2.5 mg/kg), *N*-oleoylethanolamine (OEA, 2.5 mg/kg), A8 (0.03 mg/kg) or the vehicle (DMSO) were intravitreally injected immediately after ONC while the mice were still under anesthesia. The experimenter was blinded to the identity of the compounds, including the vehicle. Before the injection of the compounds, PBS was applied to clean the cornea. A pulled glass micropipette attached to a 10 µL Hamilton syringe was used to deliver 2 µL of a solution into the vitreous chamber of the eye, posterior to the limbus. Care was taken to prevent damage to the lens. The pipette was held in place for 3 s after the injection and slowly withdrawn from the eye to prevent reflux. Injections were performed under a surgical microscope to visualize pipette entry into the vitreous chamber and to confirm the delivery of the injected solution.

### 4.4. Anterograde Labeling

On the third day prior to euthanasia at 3 weeks after the injury or within a few days after testing visual function at 12 weeks post injury, the animals were intravitreally injected with CTB-Alexa Fluor 555-conjugated cholera toxin subunit B (CTB, Life Technologies, CA, USA) as an anterograde tracer to visualize axons in the optic nerve that originated from RGCs.

### 4.5. Visual-Evoked Potentials (VEP) and Electroretinogram (ERG)

The VEP and ERG were recorded using an Espion Electrophysiology System (Diagnosys, LLC, MA, USA), as described earlier [29]. At 12 weeks after ONC, the mice were anesthetized by an intraperitoneal injection of ketamine (100 mg/kg) and xylazine (10 mg/kg). Pupils were dilated with 1% tropicamide and 2.5% phenylephrine and mice were placed on a heated platform with their head covered by the Ganzfeld dome with the uninjured eye covered. A reference needle electrode was inserted in the lower lip, while the ground electrode was inserted in the tail. For VEP recording, the test electrode was subcutaneously inserted medially on the head such that it was in contact with the skull over the visual cortex. Eyes were stimulated by the flash stimuli of white light generated by the ColorDome Ganzfeld that had an intensity of 10 cd-s/m^2^, with each set including 100 sweeps. Three sets of readings were recorded and averaged to obtain the amplitude and latency (implicit time) of the N1 component (the first negative peak: P1–N1). Standard ERG was recorded according to the photopic protocol [30], as described earlier [29]. After applying a drop of topical petroleum ophthalmic ointment, a gold wire loop electrode was placed in the center of the cornea, a reference electrode in the forehead, and a grounding electrode in the tail, and responses were recorded for 0.3 s after each stimulation. After the test, the electrodes were removed, and the mice were transferred to their home cage on a heating pad and allowed to regain consciousness and housed until further analysis.

### 4.6. Electron Microscopic Analysis

Shortly after the VEP measurements at 12 weeks after injury, the mice were anesthetized by an intraperitoneal injection of a lethal dose of ketamine/xylazine, and quickly perfused with a fixative that contained 4% paraformaldehyde, 2.5% glutaraldehyde, 0.13 N NaH_4_PO_4_, and 0.11 M NaOH, at pH 7.4. Optic nerve tissues were collected and fixed in PBS-buffered 2.5% glutaraldehyde and 0.5% osmium tetroxide, dehydrated, and embedded into Spurr’s epoxy resin [31]. Ultrathin sections (90 nm) were taken from the optic nerve approximately 100–500 µm from the legion, double-stained with uranyl acetate and lead citrate, and viewed in a JEOL JEM 1010 transmission electron microscope equipped with a digital imaging camera. EM images were taken for samples from 6 mice per each group and analyzed by NIH Image J software. The *g*-ratio (quotient axon diameter/fiber diameter) of each axon was calculated by the perimeter of axons (inner) divided by the perimeter of the corresponding fibers (outer) [32].

### 4.7. Whole Mount Retina Staining and RGC Survival Quantification

Eyes were dissected from PFA-perfused animals and left in 4% PFA for an additional hour. Whole-mount immunostaining was performed using anti-β-III tubulin (Cell signaling, 1:500) with DAPI counterstaining. Retinas were mounted onto coverslips in mounting medium (FluorSave) and imaged with a Zeiss LSM700 confocal microscope and images were acquired from 4 representative fields per retina. For each field, 4 stacked images in each perpendicular direction were acquired. The number of RGCs and neurite length were measured using Metamorph and Image J software by an observer masked to the treatment and genotype.

### 4.8. Immunohistochemistry

At 3 or 12 weeks after ONC, tissues were collected after perfusion for immunostaining. Frozen sections (25 μm thickness) were prepared using a cryostat microtome (Leica, Deer Park, IL, USA ) and fixed in 4% PBS-buffered paraformaldhyde solution, and permeabilized using 0.3% Triton-X 100 and 3% goat serum in PBS for immunostaining. Tissue samples were incubated with primary antibodies diluted in PBS that contained 3% goat serum at 4 °C overnight. The primary antibodies used were rabbit anti-GAP43 (1:500, Cell Signaling, Denve, MA, USA), anti-β-III tubulin (1:500, Cell signaling) and anti-MBP (1:500, Cell signaling). After washing 3 times with PBS, the samples were incubated with Alexa Fluor 488 (1:200, green) or Alexa Fluor 555 secondary antibody (1:200, red) (Molecular Probes, Waltham, MA, USA) for 1 h at room temperature. The samples were mounted in fluoro mounting medium (Millipore, Burlington, MA, USA) and images were taken with a Zeiss LSM700 confocal microscope (Kuehnstrasse, Hamburg, Germany). The fluorescence images were analyzed using ImageJ from the National Institute of Health (Bethesda, MD, USA) to analyze the fluorescence intensities of the tissues compared with that of the control after subtracting the background value.

### 4.9. Western Blot

WT or *gpr110 KO* mice were intracardially perfused with phosphate-buffered saline (PBS) to clear their blood before the brains were removed. Optic nerve tissues were dissected out and homogenized in ice-cold solubilization buffer (25 mM Tris pH 7.2, 150 mM NaCl, 1 mM CaCl_2_; 1 mM MgCl_2_) that contained 0.5% NP-40 (Thermo Scientific, Waltham, MA, USA) and protease inhibitors (Sigma, St. Louis, MI, USA). The protein concentrations of the lysates were determined by microBCA protein assay (Pierce, Rockford, IL, USA). The samples for SDS-PAGE were prepared at a 1 μg protein/μL concentration using 4X SDS-PAGE buffer (Lifetechnology, San Mateo, CA, USA) and 20 μg of protein was loaded onto each well. Proteins were separated by SDS-PAGE on 4–15% polyacrylamide gels (Lifetechnology, CA, USA) and transferred onto a PVDF membrane (Lifetechnology, CA, USA). After treating with a blocking buffer that contained 0.01% Tween-20, 10% BSA (Lifetechonolgy, CA, USA) for 1 h at room temperature, blots were incubated with primary antibodies diluted in blocking buffer (anti-O_2_ 1:1000, Millipore, anti-MBP 1:1000, Abcam (Waltham, MA, USA), anti-CNPase 1:1000 and rabbit anti-GAPDH 1:1000, Cell Signaling, CA, USA) overnight at 4 °C, followed by HPR-conjugated secondary antibodies (1:5000, Cell Signaling, CA, USA) for 1 h at room temperature. Detection was carried out using the KODAK Imaging System (Molecular Dynamics, Sunnyvale, CA, USA).

### 4.10. Determination of the Synaptamide and A8 Level 

Synaptamide and A8 were analyzed by reversed phase liquid chromatography coupled to high-resolution tandem mass spectrometry, using a Thermo Scientific Q-Exactive mass spectrometer as described earlier [15]. Briefly, the mice were intravitreally injected with a mixture of synaptamide and A8. At 1 and 24 h after the injection, the mice were perfused with 1× PBS, and the injected eyes were collected and homogenized in a water/methanol (1:1) mixture that contained 2 μM URB597 (a fatty acid amide hydrolase inhibitor) and 50 μg/mL butyl hydroxytoluene (BHT) (Sigma-Aldrich, St. Louis, MO, USA, cat# W218405). The homogenate was brought to BHT-methanol/water (7:3) and centrifuged for 20 min at 4 °C, after the addition of a mixture of deuterated internal standards of d_4_-synaptamide and d_6_-A8. The supernatants were loaded onto a Strata-X polymeric C18 reverse-phase SPE cartridge (33 μm, 30 mg/mL, Phenomenex, Torrance, CA, USA) that was equilibrated with water. After washing with water, the samples were eluted with 2.5 mL BHT-methanol into glass tubes, dried under N2, and resuspended in a small volume of BHT-methanol and injected onto the LC/MS/MS system. Separation was achieved using an Eclipse C18 HPLC column (1.8 μm, 2.1 mm × 50 mm, Agilent Technologies, Santa Clara, CA, USA) and a tertiary gradient consisting of water (A), methanol (B), and acetonitrile (C), with all solvents containing 0.01% acetic acid (Thermo Scientific). After pre-equilibration of the column with A/B (60%/40%), 5 μL of the extract was injected, and the solvent composition was linearly changed to A/B/C (36.3%/15%/48.7%) in 5 min, followed by a linear gradient to A/B/C (13.5%/68.4%/18.1%) over 22 min. The mass transitions of 372.3 to 62.060, 400.3 to 72.081, 376.3 to 66.085, and 406.4 to 78.118 were used to detect synaptamide, analog 8, d_4_-synaptamide, and d_6_-A8, respectively. Quantitation of synaptamide and A8 was achieved using d_4_-synaptamide and d_6_-A8 as the respective internal standards. 

### 4.11. Statistical Analysis

Data were analyzed using GraphPad Prism 7 software (Ver. 7.05). All data are presented as mean ± s.e.m. and are representative of at least two independent experiments. Statistical significance was determined by unpaired Student’s *t* test or one-way ANOVA. * *p* < 0.05, ** *p* < 0.01 and *** *p* < 0.001.

## Figures and Tables

**Figure 1 ijms-24-05340-f001:**
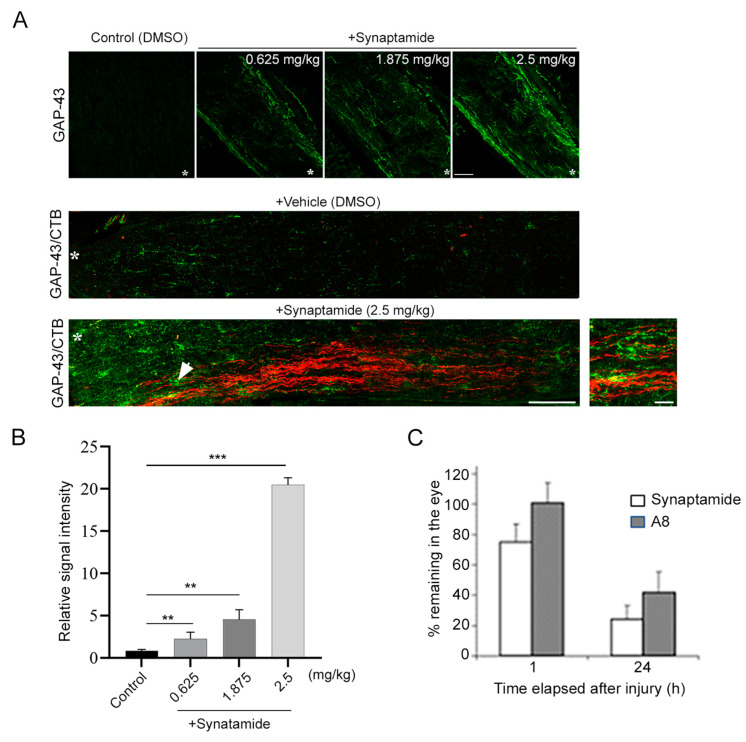
Intravitreal injection of synaptamide stimulates axon regeneration and extension after injury. (**A**). Synaptamide dose-dependent axon regeneration indicated by GAP43 staining of optic nerve (top), and synaptamide (2.5 mg/kg)-induced GAP43-positive regenerating axons (green) and axonal extension indicated by CTB labeling (red) at 3 weeks after ONC (bottom) compared to the vehicle-treated control (middle). Longitudinal sections through the optic nerve were collected at 3 weeks after ONC and regenerating axons were visualized by immunostaining for GPA-43, a regenerating axon marker (top). On the third day prior to euthanasia, the animals were injected with CTB-Alexa Fluor 555-conjugated cholera toxin subunit B (CTB, red) as an anterograde tracer to visualize axons in the optic nerve originating from RGCs (bottom). Lesion sites were marked by asterisks (*). Scale bar, 100 μm. An enlarged view around the white arrow is shown as the boxed area, indicating overlapping signals of GAP-43 and CTB. Scale bar, 20 μm. (**B**). The fluorescent intensities of GAP-43 staining shown in ((**A**) top) were quantified by setting the whole captured image as the region of interest. The signal intensity was shown relative to the control group. Data are expressed as mean ± s.e.m. (*n* = 3 per group). ** *p* < 0.01, *** *p* < 0.001. (**C**). The time course of synaptamide and A8 detected in the eye. Synaptamide (2.5 mg/kg) or A8 (0.3 mg/kg) intravitreally injected were detected by tandem mass spectrometry. The data are expressed as mean ± SD (*n* = 3).

**Figure 2 ijms-24-05340-f002:**
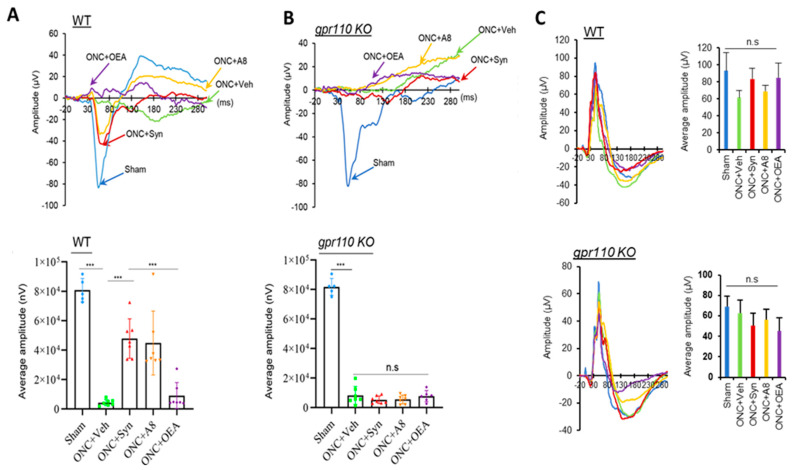
Intravitreal injection of synaptamide or A8 GPR110 dependently improved visual function impaired by ONC. (**A**,**B**), Representative VEP responses (top) and the mean amplitude (bottom) measured at 12 weeks after ONC in sham (blue), vehicle- (green), synaptamide- (red), A8-(orange) or OEA-injected (purple) C57BL/6 WT and gpr110 KO mice. ONC caused a drastic reduction in VEP compared to sham animals (83.3 ± 5.5 µV). At 12 weeks, WT mice injected with synaptamide (2.5 mg/kg) or A8 (0.03 mg/kg) showed improvement of VEP amplitude, but the gpr110 KO group did not show the same improvement. The improvement in the VEP amplitude was not shown in OEA-injected WT mice. Full-field flash VEP was elicited under light-adapted conditions (10 cd·s/m^2^). (**C**) ERG measurements at 12 weeks after ONC, showing no effects of the injury or treatment in either WT and gpr110 KO mice. Data are expressed as mean ± s.e.m. (*n* = 7), representing two independent experiments. *** *p* < 0.001. n.s., not significant.

**Figure 3 ijms-24-05340-f003:**
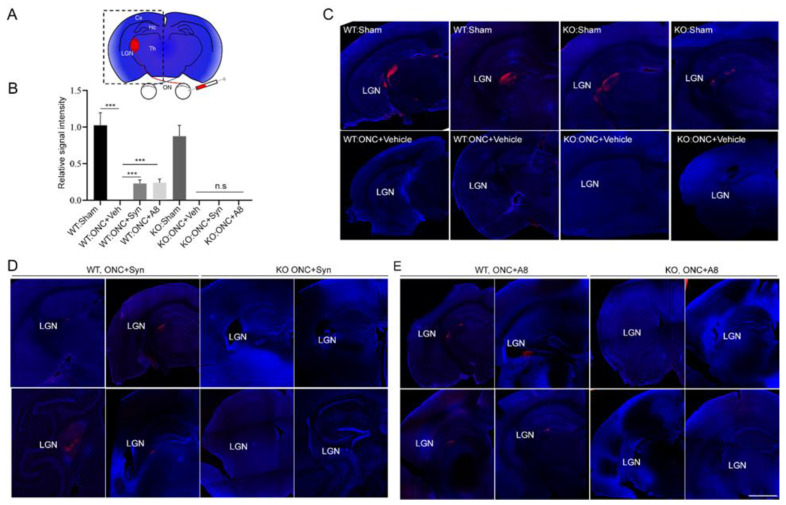
Intravitreal injection of GPR110 ligands partly prevented the ONC-induced loss of optic nerve axons at the target. (**A**) Schematic representation of a brain section depicting the visual system labeled by the CTB injection. The axons of retinal ganglion cells exit the eye via the optic nerve (ON) and project to the contralateral lateral geniculate nucleus (LGN) through the optic tract (OT). (**B**) The CTB fluorescent intensities were quantified by setting the whole captured image as the region of interest. The signal intensity was shown relative to the WT-sham group. Data are expressed as mean ± s.e.m. (*n* = 4 per group). *** *p* < 0.001. (**C**–**E**) Representative micrographs showing axon projection to the contralateral LGN region of the brain. Anterograde axon tracing of regenerated optic nerves was performed by injecting CTB conjugated to Alexa 555 on the third day prior to the scheduled euthanasia at 12 weeks after ONC. The CTB labelling that was not observed in the brains after ONC (**C**) was detected in synaptamide- or A8-injected injured WT (**D**), but not in gpr110 KO mice (**E**). The micrographs of four individual mouse brains per each group are shown with DAPI counterstaining. LGN, lateral geniculate nucleus. Cx, cortex. Th, thalamus. Hc, hippocampus. Scale bars, 100 μm. n.s., not significant.

**Figure 4 ijms-24-05340-f004:**
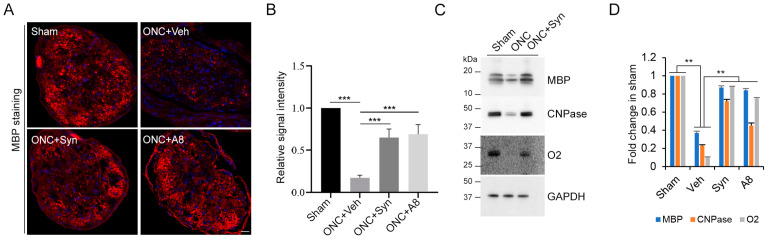
ONC-induced degradation of optic nerve myelination was partially prevented by synaptamide or A8 treatment. (**A**) Cross section of the optic nerve immunostained for MBP indicates the loss of the axon structure at 12 weeks after ONC, which is partially reversed by synaptamide or A8 injection. (**B**) The fluorescent intensities of MBP staining were quantified by setting the whole captured image as the region of interest. The signal intensity was shown relative to the sham control group. (**C**,**D**) Immunoblot of myelination marker proteins, MBP, CNPase and O2 from optic nerves collected at 12 weeks after ONC from sham and vehicle-, synaptamide- or A8-treated group (**C**), with quantitative results normalized by GAPDH from three mice per group (**D**). Data are expressed as mean ± s.e.m. (*n* = 3), representing two independent experiments. ** *p* < 0.01, *** *p* < 0.001. Scale bar, 10 μm.

**Figure 5 ijms-24-05340-f005:**
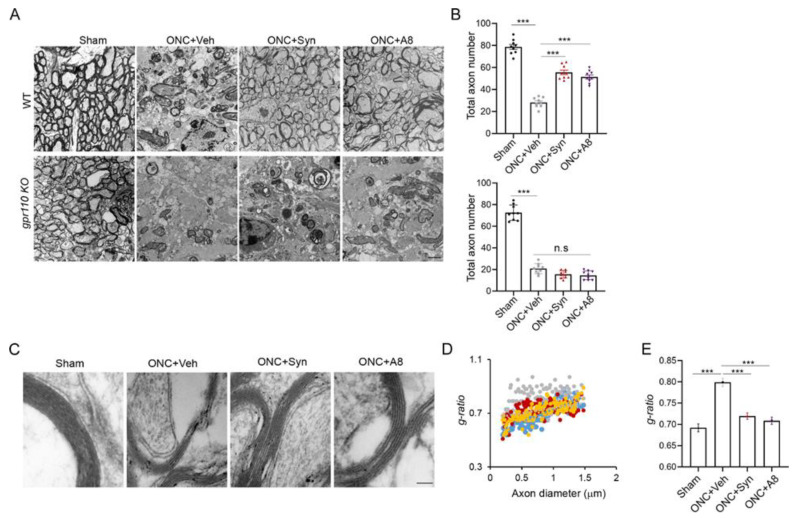
Structural integrity of axons deteriorated by ONC was improved by treatment with GPR110 ligands. (**A**,**B**) Transmission electron microscopic images show cross sections of the optic nerves from sham and vehicle-, synaptamide- or A8-treated WT or gpr110 KO mice at 12 weeks after ONC (**A**), and the quantitation of the total axon number per field (**B**). Data are expressed as mean ± s.e.m. of the total 9 sections (3 sections per mouse) from 3 mice per group. (**C**–**E**) Electron microscopic images show the thickness of the myelin sheath of the axon fibers at 12 weeks after ONC (**C**) and the scatter plot for the g-ratio (the ratio of the inner to the outer diameter of the myelin sheath) as a function of axon diameter (**D**), along with the average g-ratio for the sham (blue), ONC + vehicle (grey), ONC + synaptamide (red) and ONC + A8 (orange) groups (**E**). Data are expressed as mean ± s.e.m. of the total 114 axons from 6 mice per group. *** *p* < 0.001. Scale bars, 2 μm (**A**) and 100 nm (**C**). n.s., not significant.

**Figure 6 ijms-24-05340-f006:**
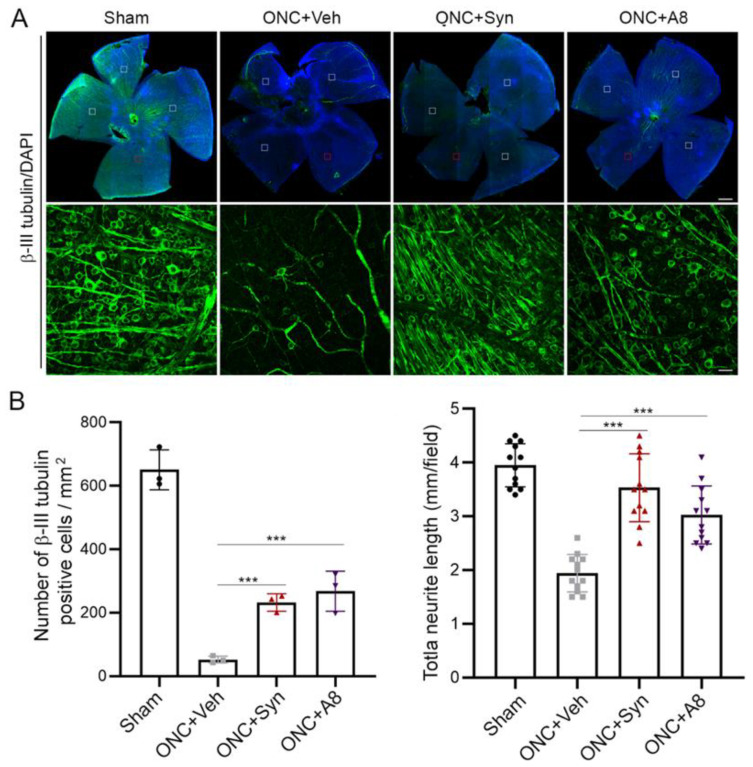
ONC-induced RGC loss was alleviated by synaptamide or A8 treatment. (**A**,**B**) Confocal images of whole mount retina where RGCs were visualized by b-III tubulin (green) and DAPI (blue) at 12 weeks post-ONC (**A**), indicating significantly greater numbers of neurons and neurites after synaptamide or A8 treatment compared to the vehicle-treated ONC control (**B**). A representative image from the red-outlined boxed region is also shown with higher magnification ((**A**), bottom). The number of b-III tubulin-positive RGC neurons and total neurite length were quantified for 4 fields (boxed regions) per retina from three animals (**B**). Scale bars in (**A**), 400 μm (top), 20 μm (bottom). *** *p* < 0.001.

## Data Availability

Not applicable.

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
