# Peer review of "Ligand-Induced Activation of GPR110 (ADGRF1) to Improve Visual Function Impaired by Optic Nerve Injury"

_ijms, 2023, doi:10.3390/ijms24065340_

Round 1

Reviewer 1 Report (Previous Reviewer 2)

The authors sent a revised version of their manuscript with some additional experiments.

- They provided a quantification of synaptamide levels in the eye by mass spectrometry. These results a quite puzzling: they show that synaptamide amount is decreased by 80% at 24h post injection. This is a very low level of compound 24h post injection compared to the extent of functional recovery they claim. It would have been interesting to quantify synaptamide levels during all the time of experiment and particularly at the time point of functional recovery test. These results would have been interesting in order to determine potential treatment and needs of several injections. These results are not compatible with the functional recovery observed.

Moreover, these results raise a new issue: Authors claim that the injury is not complete. How can they control the extent of injury in the optic nerve? Could it be that the injury was more extreme in control compared to the optic nerves treated with their drug?

- My major issue is about the figure 1A. They asses the level of GAP43 in optic nerve sections following a dose response to synaptamide. The images provided are really hard to interpret due to small portion of optic nerve and low resolution of pictures. There is also no quantification of axon regeneration or GAP43 staining. Thus, based on the images, the extend of regeneration seems to be around 100 to 200um maximum (at 2.5mg/kg) and the full circuit from the eye to the brain is about 3mm to 5mm. I cannot see how this regeneration phenotype is compatible with functional recovery. In their previous revision, authors claim that neurite sprouting is the mechanism and they justify their statement with this new figure 1A. I do not agree that GAP43 staining in optic nerve provides sufficient information to back up their claims about sprouting.

Altogether, authors failed to provide clear assessment of axon regeneration of their animals, particularly for those in functional recovery experiment. The new images of figure 1 and the very low amount of GAP43 staining makes hard to understand the effect of the synaptamide and those not provide enough strength to publish the paper as it is now.

Author Response

  1. Moreover, these results raise a new issue: Authors claim that the injury is not complete. How can they control the extent of injury in the optic nerve? Could it be that the injury was more extreme in control compared to the optic nerves treated with their drug?

Response: This is the reason why we used a blind approach with higher number of animals (n=7), which was clearly indicated in the manuscript.  We coded the drugs and thus experimenter did not know which group of animals were receiving which treatment until all experiments were completed.  This is the accepted way of conducting experiments, unless the reviewer suspects intentional manipulation of the groups on our part.  The data clearly indicated significant impact of synaptamide or A8 treatment on the recovery of axons and visual function.  We have repeated the experiments at least twice and obtained consistent results.

  1. My major issue is about the figure 1A. They asses the level of GAP43 in optic nerve sections following a dose response to synaptamide. The images provided are really hard to interpret due to small portion of optic nerve and low resolution of pictures. There is also no quantification of axon regeneration or GAP43 staining. Thus, based on the images, the extend of regeneration seems to be around 100 to 200um maximum (at 2.5mg/kg) and the full circuit from the eye to the brain is about 3mm to 5mm. I cannot see how this regeneration phenotype is compatible with functional recovery. In their previous revision, authors claim that neurite sprouting is the mechanism and they justify their statement with this new figure 1A. I do not agree that GAP43 staining in optic nerve provides sufficient information to back up their claims about sprouting.

Response: We agree that the rate of axon extension is not fast enough to cover the full length in 12 weeks.  As we mentioned in the discussion, we interpret that the observed functional recovery may be facilitated due to the repair of injured axons including sprouting of axons in an early stage of injury when axons were not yet completely degenerated.  GAP43 staining for regenerating axons is clearly indicated. In fact, we have published the similar GPA-43 images with quantitation in IJMS that as shown below (IJMS 202122 (7), 3386; https://doi.org/10.3390/ijms22073386).

Reviewer 2 Report (New Reviewer)

To analyze the effects of synaptamide and dimethylsynaptamide on nerve fiber regeneration, the authors investigated wild-type mice and gpr110 knockout mice in an experimental system of optic nerve crush.

No problems were found in the experimental methods, and the experimental results are indicative of the neuroregenerative effects of synaptamide and dimethylsynaptamide.

One minor point as below should be added.

Could synaptamide and dimethylsynaptamide be effective not only in optic nerve crush, but also in pathological conditions such as ischemia and inflammation? Please add this point to the discussion.

Author Response

Reviewer 2:

One minor point as below should be added. Could synaptamide and dimethylsynaptamide be effective not only in optic nerve crush, but also in pathological conditions such as ischemia and inflammation? Please add this point to the discussion.

As suggested by the reviewer, we have added following sentences in the discussion (page 16).

Response: It is also noteworthy that activating GPR110/cAMP signaling was shown to suppress neuroinflammation caused by traumatic brain injury [15] or endotoxin administration [32], posing possibility that these GPR110 ligands may be similarly effective in inflammation-associated neuropathological conditions such as ischemia and Alzheimer’s disease.  

Reviewer 3 Report (New Reviewer)

In the manuscript “Targeting GPR110 to Improve Visual Function Impaired by Optic Nerve Injury”, the authors investigated the protective role of GPR110 in optic nerve crush-induced RGC loss. Although the authors showed some data on axon regeneration and RGC survival improvement, I have some concerns, as listed below:

Major comments:

The title is confusing because knockout GPR110 leads to a failure of A8-induced VEP amplitude restoration. Targeting sounds like against this protein.

Quantification of Fig 1A, Fig 3, and Fig 4A is required. In addition, the CTB image seems non-specific labeling. CTB indicates regenerative axons, however, the image here looks like an optic nerve with an incomplete crush. A strong CTB signal should appear at the crush site (asterisks).

Minor comments:

Some scale bar unit is in error. Like 2 "@"m in Fig 5, 6.

Author Response

Reviewer 3: 

Major comments:

  1. The title is confusing because knockout GPR110 leads to a failure of A8-induced VEP amplitude restoration. Targeting sounds like against this protein.

Response: As suggested, we have changed the title to “Ligand-induced Activation of GPR110 (ADGRF1) to Improve Visual Function Impaired by Optic Nerve Injury”

  1. Quantification of Fig 1A, Fig 3, and Fig 4A is required.

Response: We have now added quantified data for Figs. 1A, 3, and 4A according to the reviewer’s comment. 

In addition, the CTB image seems non-specific labeling. CTB indicates regenerative axons, however, the image here looks like an optic nerve with an incomplete crush. A strong CTB signal should appear at the crush site (asterisks).

Response: With all due respect, regenerating axons are indicated by GAP-43 staining. We have previously reported GAP43-positive regenerating axons after ONC and synaptamide treatment (IJMS 2021, 22 (7), 3386; https://doi.org/10.3390/ijms22073386).

Round 2

Reviewer 3 Report (New Reviewer)

A new method for detecting regenerative axons is required to warrant the result.

Author Response

Dear Editor,

As suggested, we have included in Fig. 1A the image of control (vehicle-treated) axon stained with GAP-43 and CTB (middle panel), showing clear difference in staining intensity observed from synaptamide (2.5 mg/kg)-treated axons (bottom).  

We sincerely hope that our response properly addresses the reviewer's concern and the manuscript is now acceptable for publication.    Thank you.  

This manuscript is a resubmission of an earlier submission. The following is a list of the peer review reports and author responses from that submission.

Round 1

Reviewer 1 Report

This is an interesting paper investigating the role of GPR110 agonist in a mouse model of optic nerve injury. This model has been investigated for a long time and allows to discriminate between neuronal survival ( ganglion cells) and axonal regeneration. After optic nerve crush axons degenerate and ganglion cells die. It is not clear in the paper whether the authors were able to differentiate the mechanism of action of the GPR110 agonists. The methods are well described however the number of animals used in the experiments is not shown ( In some Figure legends they are). The Results section also needs some improvement. For instance, in Fig . 2 it is not possible to see any CT labelling in the experimental groups (only in the sham ).   It is also not clear how they quantified for instance the number of axons in Figure 4 A and B  (Total axon number ? In each field? In each nerve ? 100 axons/nerve ?). 

It would also be interesting to describe better the gpr110KO phenotype ( the visual system ). Do they have the same number of axons as the WT ( the sham groups?). If the GPR110 receptor is so important for axonal outgrowth ( and regeneration) should not they have a smaller number of axons? Some mechanisms involved in the development of the visual system are also involved in regeneration.

Fig . 5. Ganglion cell number : in each retina for fields were quantified ( although only 3 are represented) . Usually the number of ganglion cells should be represented /mm2 of retina or the total number of cells per retina . In this case it also would be interesting to know the number of ganglion cells in the grp110KO animal (since this animals were also used in the functional assays). 

Fig 5B : Total neurite lenght(mm/mm2) : I do not understand what the authors are actually evaluating . Neurites in the retina flat mounts with betaIII -Tubulin labeling ? In our experience most of the "fibers" seem in flat mounted retinas with this labeling are actually axons and in this case usually these "regenerated" axons are best identified in the nerve (after the crushed regions with for instance CT labeling ).

Author Response

This is an interesting paper investigating the role of GPR110 agonist in a mouse model of optic nerve injury. This model has been investigated for a long time and allows to discriminate between neuronal survival (ganglion cells) and axonal regeneration. After optic nerve crush axons degenerate and ganglion cells die. It is not clear in the paper whether the authors were able to differentiate the mechanism of action of the GPR110 agonists.

Response: We have previously demonstrated that these GPR110 agonists transmit the cAMP signaling and promote the nerite growth (Lee et al., 2016; Kwon et al., 2021).  However, further studies are required to delineate the mechanism leading to axonal growth and improvement of RCG survival as mentioned in the discussion. 

The methods are well described however the number of animals used in the experiments is not shown (In some Figure legends they are).

Response: We now indicate n numbers in each figure legend, wherever appropriate.

The Results section also needs some improvement. For instance, in Fig. 2 it is not possible to see any CT labelling in the experimental groups (only in the sham).   

Response: Although the labeling is indeed low due to partial recovery, the presence of CTB labelled axons is evident.

It is also not clear how they quantified for instance the number of axons in Figure 4 A and B (Total axon number? In each field? In each nerve ? 100 axons/nerve ?). 

Response: In figure 4, the axons were counted per each field, and this is now indicated in the figure legend.

It would also be interesting to describe better the gpr110KO phenotype (the visual system). Do they have the same number of axons as the WT (the sham groups?). If the GPR110 receptor is so important for axonal outgrowth (and regeneration) should not they have a smaller number of axons? Some mechanisms involved in the development of the visual system are also involved in regeneration.

Response: Given that GPR110 is important for developmental neurite outgrowth, GPR110 may influence RGC axonal outgrowth during development.  Nevertheless, in the long run, other mechanisms for axonal outgrowth can compensate the developmental deficit from the lack of GPR110 activation.  Indeed, the total RGC axon number per field (Fig. 4A,B) as well as the VEP amplitude (Fig.1) appeared similar between WT and KO sham animals. On the other hand, GPR110 activation after injury can stimulate the neurite growth mechanism leading to the timely repair of axons and the improvement of visual function.

Following sentences have been incorporated in the second paragraph of the discussion.

Given that GPR110 plays an important role in developmental neurite outgrowth, lack of GPR110 may have affected RGC axonal outgrowth during development.  However, the total RGC axon number per field (Fig. 4A,B) as well as the VEP amplitude (Fig.1) appeared similar between WT and KO sham animals, suggesting that unlike the injury situation, other mechanisms for axonal outgrowth can compensate in the long run for the developmental deficit caused by the lack of GPR110 activation. 

Fig . 5. Ganglion cell number : in each retina for fields were quantified (although only 3 are represented) . Usually the number of ganglion cells should be represented /mm2 of retina or the total number of cells per retina. In this case it also would be interesting to know the number of ganglion cells in the grp110KO animal (since this animals were also used in the functional assays). 

Response: The preliminary test with WT and KO Sham animals showed similar RGC numbers (data not shown). As the total RGC axon numbers were similar between WT and KO Sham animals and the recovery was GPR110-dependent (Fig. 4A,B), we only evaluated the effect of ligands on RGC survival in WT animals.  Obviously, further studies will be required to fully understand the GPR110-derived trophic signals for RCG survival after injury. 

Fig 5B : Total neurite length (mm/mm2) : I do not understand what the authors are actually evaluating . Neurites in the retina flat mounts with betaIII -Tubulin labeling? In our experience most of the "fibers" seem in flat mounted retinas with this labeling are actually axons and in this case usually these "regenerated" axons are best identified in the nerve (after the crushed regions with for instance CT labeling ).

Response: The purpose of the beta-III tubulin staining in Fig. 5 is to demonstrate the effects of the treatment with GPR110 ligands on RGC survival after injury. The RGCs status was indicated by the beta-III staining of RGCs in the cytoplasm, dendrites and axons in each condition.  The data showed that the staining of beta-III tubulin positive neurons (RGC) as well as neurites decreased significantly after injury but partially recovered by the treatment with GPR110 ligands.  The data support the improved RGC survival status after the treatment with GPR110 ligands.

Reviewer 2 Report

In this study Authors aim to decipher visual improvement after optic nerve injury by targeting GPR110. They first test potential functional recovery by performing VEP test. They also analyzed CTB signal in visual brain target (LGN) to see extend of axon regrow. In parallel, the author looked at axonal integrity by doing immunofluorescence against myelin protein and performed electron microscopy to check myelin sheet. Finally, the author looked at the level of neuroprotection induced by their treatment. Altogether, the authors claims that their treatment induce circuit formation and functional recovery. However, the study lacks the appropriate and key controls in order to be fully convincing and be published.

Specific comments:

- In material and methods, the age of mice is not clear.

The concentration of drugs is given in g/kg: however, most of the injections are intravitrally. This could be confusing. Authors should state clearly how much of the drugs are injected. It is not clear how long authors waited before injecting the CTB.

- For the VEP experiments, were both optic nerves injured?

- Section 2.6 does not match with the last sentence of the sections 2: in 2.6 it is stated that mice undergo intracardiac perfusion right after the VEP experiment. In Section 2.5 mice are allow to recover.

Please make the materiel and methods consistent.

 Several critical controls and information are missing:

* is the optic nerve crush unilateral?

* what is the read out of drug treatment?

* how the doses of drugs have been determined?

* what is the half-life of the drugs?

* is the injury complete in all animals?

*what is the extent of regeneration in the optic nerve for all animals used for the VEP experiments and electronic microscopy?

- Results presented in the group’s previous publication (Kwon et al 2021) show little regeneration of the optic nerve upon using the same experimental paradigm. Therefore, it is not understandable how there is functional recovery and LGN innervation? It is widely known in the field of axon regeneration that regenerative axons in the optic nerve tend to die back after 4 weeks post injury.

Indeed, in figure 1, authors showed results from VEP experiment with recording of activity for synaptamide and A8 treatment. However, they do not present any picture of optic nerve section (or full optic nerve transparency) to assess the level of RGC regeneration and the quality of the optic nerve injury. this statement is true for all the other conclusions of the article. How to define the extend of functional recovery when we cannot measure regeneration in the optic nerve. The author presented CTB staining in LGN in figure 2. How the author can be sure it is not spared axons still present after optic nerve crush? Without a clear demonstration of the completeness of the crush, the author cannot conclude clearly on the effect of the treatment.

- In figure 4, the author shows myelin sheet around RGC axon after optic nerve crush. Does

synaptamide and A8 treatment promote myelination of regenerative axons? It is a surprising result that need to be more clearly demonstrated. It is known that ONC induce total degradation of axon post injury site and regenerative axons are not myelinated except after certain manipulation (Wang et al, Neuron 2020). As stated before, it looks like more spared axons left intact by incomplete ONC. How injection of the drug in the eye could modulate oligodendrocytes in the optic nerve?

From the picutures presented in fig 5A, it appears that Tuj1 staining is not homogenous in all the retina. These technical issues could induce an experimental bias, making the results hard to interpret or to be convincing.

Other comment:

-       The introduction is relatively brief. It would be helpful for the reader to have a better contextualization of the work with better description of what is known. Particularly all the works that has bee, done on axon regeneration and functional recovery from Zhigang He, Benowitz, Goldberg, Park or Hubermann among other.

Author Response

Specific comments:

- In material and methods, the age of mice is not clear.

Response: The mice were 2 month-old.

The concentration of drugs is given in mg/kg: however, most of the injections are intravitrally. This could be confusing. Authors should state clearly how much of the drugs are injected. It is not clear how long authors waited before injecting the CTB.

Response: Considering that the weight of the mice was about 20 g, each mouse was injected 2.5 x 20=50 (synaptamide) and 0.03 x 20=0.6 (A8) microgram, although the exact amount injected was according to the weight of each mouse.  The CTB injection was performed at 3 days prior to the harvesting within a few days after VEP measurements at 12 weeks post injury.

- For the VEP experiments, were both optic nerves injured?

Response: Only one eye was injured, and for VEP measurement, the uninjured eye was covered, which is indicated now in the experimental section.

- Section 2.6 does not match with the last sentence of the sections 2: in 2.6 it is stated that mice undergo intracardiac perfusion right after the VEP experiment. In Section 2.5 mice are allow to recover. Please make the materiel and methods consistent.

Response: We have added the phrase ‘and housed until further analysis’ at the end of the section 2.5 to be consistent.

 Several critical controls and information are missing:

* is the optic nerve crush unilateral? Yes.

* what is the read out of drug treatment?  We measured VEP, axon extension, axon numbers and RGC survival as the readout. 

* how the doses of drugs have been determined?

Response: In our previous study using the same injury model, we examined the effects of the several different doses of the drugs for the regeneration of axons using anti-GAP43 staining to determine the dose (Kwon et al., 2021).

* what is the half-life of the drugs?

Response: The half-life of the drugs is 6-12 h.  Nevertheless, a single intravitreal injection of 2.5 mg/kg synaptamide appears sufficient to sustain enough local concentration to promote the axon regeneration and repair at the initial stage in WT animals.

* is the injury complete in all animals?

Response: As mentioned in the discussion, axonal injury caused by ONC is known to be mild. It has been reported that RGC survival of greater than 40% can be achieved depending on the severity of crush. We exposed the optic nerve and grasped approximately 1-3 mm from the globe with cross-action forceps for 3 s to apply the pressure on the nerve. Incomplete injury is indicated by more than 20% neurons that were spared at 12 weeks after injury (Fig. 5) along with some axons detected from the optic nerve after the crush injury (Fig. 4).

*what is the extent of regeneration in the optic nerve for all animals used for the VEP experiments and electronic microscopy?

Response: Partial restoration of axons after drug treatment was apparent from the data in Fig. 4 obtained from the optic nerve of approximately 100-500 µm length from the legion, and also from the appearance of CTB-labeled RGC axons in the LGN area (Figure 2).  However, it is not clear to what extent the restoration was contributed by axons spared due to the repair at an early stage of injury or by the extension of the regenerating axons. 

- Results presented in the group’s previous publication (Kwon et al 2021) show little regeneration of the optic nerve upon using the same experimental paradigm. Therefore, it is not understandable how there is functional recovery and LGN innervation? It is widely known in the field of axon regeneration that regenerative axons in the optic nerve tend to die back after 4 weeks post injury.

Indeed, in figure 1, authors showed results from VEP experiment with recording of activity for synaptamide and A8 treatment. However, they do not present any picture of optic nerve section (or full optic nerve transparency) to assess the level of RGC regeneration and the quality of the optic nerve injury. this statement is true for all the other conclusions of the article. How to define the extend of functional recovery when we cannot measure regeneration in the optic nerve. The author presented CTB staining in LGN in figure 2. How the author can be sure it is not spared axons still present after optic nerve crush? Without a clear demonstration of the completeness of the crush, the author cannot conclude clearly on the effect of the treatment.

Response: As mentioned above, the mechanism for the restoration of RGC axons in LGN or functional recovery after injury is not clear.  We agree that extension of regenerating axons may not be the major cause for the restoration effects, based on the extent of axon growth observed in our previous study.  Nevertheless, stimulated neurite growth by GPR110 activation may lead to the repair of injured axons at an early stage, prevent neuronal cell death, and promote the capacity to sprout neurites to form new synapses for signal transmission leading to the functional recovery. In such case, we can state that some neurons were spared by the treatment.  As the identity of the treatments was blinded to the experimenter, the observed outcome of improved functional recovery was not due to the spared axons from irregular experimental procedure, but due to the spared axons and neurons specifically by the treatment with GPR110 ligands. 

- In figure 4, the author shows myelin sheet around RGC axon after optic nerve crush. Does

synaptamide and A8 treatment promote myelination of regenerative axons? It is a surprising result that need to be more clearly demonstrated. It is known that ONC induce total degradation of axon post injury site and regenerative axons are not myelinated except after certain manipulation (Wang et al, Neuron 2020). As stated before, it looks like more spared axons left intact by incomplete ONC. How injection of the drug in the eye could modulate oligodendrocytes in the optic nerve?

Response: Again, we can speculate that effects on the myelination status may also be due to timely repair of axons by GPR110 ligands. 

From the picutures presented in fig 5A, it appears that Tuj1 staining is not homogenous in all the retina. These technical issues could induce an experimental bias, making the results hard to interpret or to be convincing.

Response: The stitching process from confocal z-stack images caused the appearance of uneven staining, which did not hinder the counting of the neurons or neurite length. 

Other comment:

The introduction is relatively brief. It would be helpful for the reader to have a better contextualization of the work with better description of what is known. Particularly all the works that has been done on axon regeneration and functional recovery from Zhigang He, Benowitz, Goldberg, Park or Hubermann among other.

Response: We have added some references in the introduction.  The work of He et al. and Park et al. has been already cited in the original manuscript.

Round 2

Reviewer 2 Report

The author submitted a revised version of their manuscript. Some minor points have been addressed (in method section). However, major concerns have not been addressed. Some critical controls are still missing.

- The read out of the drugs treatment is not controlled properly. Looking at neuroprotection or regeneration cannot be considered of a correct readout.

- Giving the functional recovery results and the literature available, it is really odd that a single injection of the drug with a so short half-life is sufficient. Again, controlled of drugs effect over time would be necessary to back up the finding.

- Author state that ONC is mild with more than 40% TGC survival. This statement is in contradiction with major work of the field (Work from Zhigang He, Larry Benowitz, Kevin Park, Josh Sanes to cite a few). ONC has been widely used and several studies demonstrated that ONC is complete with no spare axons and RGC survival less than 20%. Incomplete crush makes everything hard to interpret. How to discriminate between spare circuit and drug effect.

- author suggest that sprouting of neurite is the mechanism for recovery. However, no data is available to justify this statement.

For all this reason I cannot endorse the study for publication.